# Complementary Feeding Practices: Recommendations of Pediatricians for Infants with and without Allergy Risk

**DOI:** 10.3390/nu16020239

**Published:** 2024-01-12

**Authors:** Emilia Vassilopoulou, Gavriela Feketea, Ioannis Pagkalos, Dimitrios Rallis, Gregorio Paolo Milani, Carlo Agostoni, Nikolaos Douladiris, John Lakoumentas, Evangelia Stefanaki, Zenon Efthymiou, Sophia Tsabouri

**Affiliations:** 1Department of Nutritional Sciences and Dietetics, International Hellenic University, 57400 Thessaloniki, Greece; vassilopoulouemilia@gmail.com (E.V.); ipagkalos@ihu.gr (I.P.); john.lakoo@gmail.com (J.L.); zenonefthy@gmail.com (Z.E.); 2Pediatric Unit, Fondazione IRCCS Ca’ Granda Ospedale Maggiore Policlinico, 20122 Milan, Italy; gregorio.milani@policlinico.mi.it (G.P.M.); carlo.agostoni@policlinico.mi.it (C.A.); 3Department of Pediatrics, “Karamandaneio” Children’s Hospital of Patra, 26331 Patras, Greece; 4Department of Pharmacology, “luliu Hatieganu” University of Medicine and Pharmacy, 400349 Cluj-Napoca, Romania; 5School of Health Sciences, University of Ioannina, 45110 Ioannina, Greece; drallis@uoi.gr (D.R.); stsabouri@gmail.com (S.T.); 6Department of Clinical Sciences and Community Health, Università degli Studi di Milano, 20157 Milan, Italy; 7Allergy Unit, 2nd Pediatric Clinic, University of Athens, 11527 Athens, Greece; ndouladiris@gmail.com; 8Department of Pediatrics, General Hospital of Heraklion, Venizeleio and Pananio, 71409 Heraklion, Greece; linastef74@gmail.com

**Keywords:** allergy prevention, complementary feeding, food allergens, healthy growth, weaning

## Abstract

Aim: To investigate the routine guidance provided by pediatricians concerning the timing of complementary feeding (CF) for both healthy infants and those at a heightened risk of allergies. Methods: A total of 233 pediatricians participated in an anonymous online survey that included questions about demographics and recommendations for CF. Specifically, they provided guidance on the types of foods, preparation methods, supplements, time intervals for introducing new foods to infants at low and high allergy risk, and delayed food introductions for high-risk cases. Results: The respondents advised introducing certain foods at specific ages: fruits, starchy non-gluten grains, vegetables, olive oil, and meat were appropriate at 6 months; gluten-rich grains at 7 months; yogurt, hard-boiled eggs, and legumes at 8 months; fish at 8.5 months; and nuts at 9 months. Pediatricians, especially those with less than 15 years of practice, often introduced egg, seafood, gluten-rich grains, legumes, and nuts earlier for high-risk infants. Parenthood and male gender were associated with the earlier introduction of eggs and grains. Conclusions: Greek pediatricians follow a structured food introduction schedule for CF in infants. Interestingly, they tend to delay the introduction of common food allergens and recommend longer intervals between introducing new foods, particularly for high-risk infants. Key Notes: Despite recent evidence-based indications on healthy complementary feeding strategies for infants, discrepancies persist among pediatricians regarding food choices and the order and timing of food introduction, both for healthy infants and those at risk of allergy. Guidance on complementary feeding by pediatricians is influenced by their individual characteristics. Pediatricians tend to delay the introduction of common food allergens and recommend longer intervals between introducing new foods, particularly for high-risk infants.

## 1. Introduction

For infants, the transition from a milk-based diet to solid foods is important for healthy psychosomatic development [1,2,3,4,5] and plays a determining role in the future dietary variety of the child [6,7].

Complementary feeding (CF) in exclusively breastfed infants is recommended at approximately 6 months. In formula-fed infants, this should begin not before 17 weeks and not later than 26 weeks of age [4,8].

Delayed introduction of allergenic foods was recommended in the past based on the theory that the immature gut is more permeable, allowing food proteins to enter the bloodstream intact and potentially increasing the risk of developments of food allergy (FA) [9]. A recent shift towards earlier introduction, from 4 months onwards [10], followed the results of large-scale interventional studies in the general population [11] and in infants at high risk for allergy [12,13,14,15,16,17], although the results of both observational and interventional studies are conflicting [18,19,20,21].

Consequently, the specific time point for initiating CF remains an individual decision of pediatricians, one based on the overall development of the infant and the personal and family history of allergy [22,23,24,25,26].

Pronounced differences between pediatricians are reported regarding their guidance in terms of allergy prevention for both healthy infants and those at risk of allergy [27]. The period they recommend between the introduction of two new foods parallels the personal convictions of the pediatricians about allergy prevention, ranging from 2–3 days for healthy infants to more than 4 days for infants considered at high risk of allergy [1,2]. The Canadian Pediatric Society (CPS) recommends a flexible approach in which foods not included in the list of main allergens are introduced freely, with a time interval of a few days between new foods only observed for food allergens. Allergen-related foods should be introduced one at a time to identify more easily the causative food of a possible allergic reaction [28].

In this study, we aimed to investigate (a) the routine CF guidance for healthy infants provided by pediatricians practicing in different areas in Greece, (b) the differences in their recommendations for infants at high risk of allergy in terms of food introduction order, and (c) their recommended time period between the introduction of new foods.

## 2. Methods

### 2.1. Participants

Greek pediatricians registered with the Hellenic Pediatric Society (n~2000 members) were invited via e-mail to participate in this study from May to December 2022. A detailed introductory informative text regarding the scope of the study was provided, and online informed consent was provided by each participant. After this, they accessed the online study questionnaire using a personalized code.

The study was approved by the Education Department of the 6th Health Region of Greece (study ID 3111/18.03.2022) and conducted in accordance with the code of Ethics of the World Medical Association Declaration of Helsinki.

### 2.2. Questionnaire

A questionnaire was developed for the purposes of the study ([link to Appendix A]) and responses were gathered online using a purpose-built web platform, hosted by the International Hellenic University, which secured on-premises collection and analysis of data. For inclusion in the analysis, pediatricians were required to confirm ongoing or completed pediatric training, and to complete the questionnaire up to and including the final question.

The questionnaire included anonymous demographic information on the participant: sex, age, pediatric subspecialty, years of experience, place of work, marital status, and parental status.

The main questionnaire was divided into three sections with user-friendly multi-part ranking questions. In these, participants were asked to select foods from each food category (Section 1), as well as different cooking methods (Section 2) and food supplements (Section 3), and arrange them in the order that they usually recommend for introduction into the infant’s diet, proceeding according to month of age, given from birth to 18 months.

Each section was followed by an open-ended question for adding food items, cooking methods and food supplements not included in the list, respectively.

The time period between the introduction of new foods was explored for infants at both low and high risk of allergies, and an open-ended question recorded the foods for which introduction was delayed for infants at high risk of allergy. Completion of all sections and questions of the questionnaire was mandatory.

The questionnaire was piloted with a small number of pediatricians (n = 23) to ensure that the requested information was clearly formulated and that the responses were appropriate. The average time required for completion, based on the piloting data, was 18 min.

### 2.3. Statistical Analysis

The normality of distribution of continuous variables was assessed via the Kolmogorov–Smirnov test. Continuous variables were expressed as median (interquartile range), and comparisons were conducted using the non-parametric Mann–Whitney U test. Categorical variables were expressed as n (percentage %) and compared using the chi-square test or Fisher’s exact test. Univariate analysis was conducted on respondents’ dichotomous categorical data: sex (male, female), years of experience (<15 years, ≥15 years), location of practice (urban, rural/semi-urban), parenthood (yes, no), and subspecialty (yes, no), and the time of introduction (months of age) of each food.

Multivariate logistic regression analysis was utilized to explore the association of the characteristics of the pediatricians with the time (early, late) of food introduction and the time period between the introduction of new foods for infants at low risk and high risk of allergy. Foods introduced late into the diet of infants at high risk of allergy were examined in the models, as follows: >7 months for wheat products and trahanas (frumenty); >8 months for hard-boiled egg, cream cheese, tomato, orange, kiwi or strawberry; >9 months for soft egg/omelet/fried egg; 10 months for cows-milk yogurt; and 12 months for all other foods [12,29,30,31]. The time period between introduction of new foods was considered low if ≤3 days and high if >4 days [27].

All tests were two-sided, and a *p*-value of <0.05 was considered statistically significant (alpha = 0.05). The data were analyzed using SPSS Statistics (IBM SPSS Statistics for Windows, Version 24.0, Armonk, NY, USA).

## 3. Results

### 3.1. Participants

Of the ~2000 pediatricians invited to participate, 348 submitted responses (response rate 17.4%), of which 233 questionnaires were fully completed and were included in the final data analysis (11.65%). Of the final participants, 75.96% were female, 44.63% were based in an urban area, and 85.83% were parents; 51.9% had working experience as pediatricians of ≥15 years, and a pediatric subspecialty had been attained by 25.32% (Appendix A).

### 3.2. Infant Feeding Practices: The Order of Introduction of Solid Food Recommended by the Pediatricians

Figure 1 and Appendix A present the responses of the pediatricians regarding the usual guidance for the CF of infants that they provide to parents. Fruits are the first food to be introduced into the infant’s diet, starting at 5 months with apple and pear, followed by banana, apricot, peach, and nectarine, and then by orange, melon, watermelon, and kiwi. Grapes, pomegranate, fresh berries, and figs follow, while strawberries, dried fruits and fresh juices are suggested after 8 months (Figure 1).

Starchy gluten-free products are recommended at 5–6 months in the form of potato, white rice and gluten-free oats, followed by oats without a gluten-free indication, whole-grain rice, peas, and corn. White-wheat (gluten-rich) products are recommended at 6–7 months, and whole-wheat products and trahanas (frumenty) at 6–8 months.

A variety of vegetables, including carrot, zucchini, cabbage, spinach, green leafy vegetables, cauliflower and beetroot, are introduced at 5–6 months, and peppers, tomato, and eggplant are included slightly later.

Olive oil is introduced at 5–6 months, and olives, seed oils and margarine after 12 months. Regarding animal products, chicken and beef are recommended at 6 months, followed by lamb and rabbit. Then liver, hare, wild boar and game birds, and finally pork, are included at 7–12 months, and processed meats (cold cuts, salami and ham) are suggested at 12–18 months.

Regarding dairy products, pediatricians suggest kids’ yogurt at 6–8 months, traditional or sheep’s or cow’s milk yogurt at 7–12 months, manouri (soft) cheese at 7.25–12 months, cottage cheese at 8–12 months, and other cheese types and butter after 12 months.

Hard-boiled egg is introduced at 8 months, while other forms of egg follow later, with soft egg/omelet/fried egg and egg-lemon sauce at 9–12 months and raw egg at 11.5–18 months.

Among legumes, lentils and chickpeas are introduced first (7–10 months), with fava and regular beans slightly later.

Regarding fish, hake is introduced at 7–10 months, followed by other fish species. Shellfish are suggested after the 12th month, with canned tuna at 12–18 months.

Nuts are introduced first in the form of almond, peanut, or hazelnut butter at 7–12 months, followed by ground nuts and seeds, such as almonds, walnuts, sesame, pistachios, sunflower/pumpkin seeds, and peanuts, after 10 months.

Regarding cooking and preparation methods, the pediatricians recommend that food should be consumed first boiled and mashed at 5–6 months, followed by fork-mashed food at 8–10 months, and food in pieces at 9–12 months. Baby-led weaning (BLW) is recommended by some at 6–12 months. Sharing family food without salt is suggested at 9–12 months, and with salt at 12–18 months, and with roasted or fried food, salt and spices included in the infant’s diet after 12 months, in addition to honey, sugar and other sweet substances.

### 3.3. The Influence of Years of Pediatric Practice on the Order of Food Introduction

Pediatricians with more years of practice (≥15 years) introduce certain food items, including raisins, fresh mixed fruit juice and white rice, and food supplements, such as omega-3, vitamin C and multivitamins, earlier. Conversely, pediatricians with <15 years of practice introduce earlier various types of meat products (rabbit, beef and pork), soft and raw egg, cow’s milk and yogurt, legumes, and various fish and shellfish species (anchovy, calamari and octopus) (Table 1).

### 3.4. The Influence of Sex of the Pediatrician on the Order of Food Introduction

Male pediatricians recommend the earlier introduction of certain fruit (kiwi, pomegranate and raisins), dairy products (yogurt, Greek soft cheese and ariani), and baby biscuits with sugar. Female pediatricians suggest the earlier introduction of soft egg, olive oil and boiled and mashed food into the infant’s diet (Table 2).

### 3.5. The Influence of the Area of Practice on the Order of Food Introduction

Pediatricians practicing in rural and semi-urban areas introduce rabbit, ariani, hazelnut, and iron supplements earlier, while they delay the introduction of hard yellow cheese (kasseri/graviera/kefalotyri) in comparison to those practicing in urban areas (Appendix A).

### 3.6. The Influence of Parenthood on the Order of Food Introduction

Pediatricians with children of their own are more likely to recommend the earlier introduction of certain fruits, such as cherries and fresh orange juice, whole-wheat and white-wheat products and fork-mashed food, but to delay BLW and the introduction of cottage cheese, in comparison to non-parents (Appendix A).

### 3.7. Differences in Recommendations of Pediatricians on Food Introduction, according to High and Low Risk of Allergy

For infants at high risk of allergy, some pediatricians reported that they recommend delaying the introduction of certain items, specifically, orange (16%), kiwi (21%), strawberry (43%), gluten-rich wheat products (12%), milk and dairy products (49%), all forms of egg (59%), peanuts (57%), tree-nuts (62%), legumes (62%), fish and shellfish (48%), beef (0.4%), and soya (0.4%).

Regression models, created to explore the characteristics of the pediatricians that may predict the earlier introduction of the above-mentioned foods, showed that a shorter period of practice in pediatrics, lack of a subspecialty, parenthood and sex all differentiated the age of recommended introduction of the above foods (Table 3).

### 3.8. Factors Associated with Recommendation of a Longer Time Period between Introduction of New Foods

Applying the regression model on the factors influencing the recommended time period between the introduction of new foods (Table 4) showed that male pediatricians recommended a longer time period between introduction of new foods (>4 days) in infants at low risk of allergy. For infants at high risk of allergy, a lack of a subspecialty and practicing in a rural/semi-urban area were predictors of a longer time period between the introduction of new foods, as shown in Table 4.

## 4. Discussion

Knowledge accumulated from a variety of studies provides evidence of the benefits of healthy CF strategies in infancy for the promotion of balanced, lifelong dietary habits [32], with avoidance of obesity [33] and malnutrition [34], and a reduction in food allergies (FAs) [35,36]. In several countries, including Greece, practicing pediatricians play an important role in choosing the most suitable CF approach for guiding parents in the introduction of new foods into their infants’ diets [25]. This study is the first to our knowledge to evaluate the typical guidance on CF strategies provided by practicing pediatricians in relation to their individual characteristics. The findings reveal that the recommendations of the Greek pediatricians show similarities with the Italian infant feeding practices [37,38], in proposing that fruits, cereals, and vegetables should be introduced first. Conversely, in a US study, cereals were reported to be the first CF [27]. Notably, the Greek pediatricians show variation, and according to their individual characteristics, they may delay the introduction of several common food allergens into the diet of infants at both low and high risk of allergy and may increase the time period between the introduction of new foods for infants at high risk of allergy.

The variations do not always conform with current guidelines; specifically, Greek pediatricians suggest the introduction of gluten-containing products at around the 7th month, although the WHO guidelines [1,2,3] recommend inclusion between the 4th and the 7th months in small portions. It is also proposed that breastfed infants should continue breastfeeding for at least 2–3 weeks [8,39], while gluten consumption must be continuous up to the 1st year, in predisposed individuals [4], to reduce the risk of celiac disease [40,41], although one randomized trial suggested that the late introduction of gluten-rich foods might not increase the risk of celiac disease [41].

Among the gluten-rich foods, Greek pediatricians recommend introduction into the infant diet of trahanas (frumenty), a traditional fermented food produced from cracked wheat and milk [42]. The fermentation of dairy products has been proposed as a method of reducing allergenicity [43] and has been used effectively to increase tolerance in children with milk allergy [44]. Koksal and colleagues have proposed that pregnant and lactating mothers should consume trahanas and other fermented dairy products to protect their infants from cow’s milk allergy [45]. Greek pediatricians practicing in urban areas tended to delay slightly the introduction of other fermented dairy products, such as traditional yogurt, suggesting their introduction at around the 1st year of age for low-risk infants, or even later for infants at high risk of allergy.

Pediatricians practicing in rural/semi-urban areas recommend the earlier introduction of local dairy products, such as ariani—a fermentation product known to benefit the gut microflora and the immune system [46]. This group of pediatricians also recommends the early introduction of meat from domestic animals, such as rabbit, and game meat, as well as sesame, tree nuts, seafood, legumes and tomato, probably because of their local availability [47,48]. Residency in rural/semi-urban environments has thus been proposed as a factor that increases adherence to traditional dietary patterns and provides wider food diversity in comparison to urban environments [49].

Practicing in rural/semi-urban areas, however, along with lack of a subspecialty, were correlated with recommendation by the pediatricians of a longer time period between the introduction of new foods for infants at high risk for allergy. Studies on extended time periods between new foods suggested by pediatricians have emphasized the possible detrimental effects of delays in food introduction on dietary diversity in infancy [27], and its possible relationship with an increased risk of pediatric asthma and allergies [50,51,52,53].

Greek pediatricians in this study recommend the early introduction of sources of highly bioavailable iron into the infants’ diet at 6 months, such as red- and white-meat products. These are crucial to covering their iron needs [3] and preventing iron deficiency in exclusively breastfed infants [54].

They suggest delaying the introduction of ultra-processed meats, such as cold cuts and salami, until 18 months. Increased consumption of ultra-processed foods (UPF) has been correlated with wheezing [55], raised IgE levels, current asthma, and eczema during childhood [56]. In their responses, the female pediatricians paid more attention to food processing and cooking methods, while the male professionals suggested earlier introduction of fresh fruits and dairy products. In a relevant US study, male pediatricians provided more specific guidance than females to parents regarding healthy eating, including avoidance of sugar, enhancing fruit and vegetable variety, and exposure to many tastes and textures, while the females focused more on overall healthy eating and living [57].

Male sex emerged as a predictor for recommendation of longer time periods (>4 days) between new foods for infants at low risk of allergy. In line with our findings, female pediatricians [58] and those with less years of practice appeared, in previous studies, better informed regarding the implementation of guidelines on the appropriate timing of introduction of known allergens to prevent food allergy [59]. In this study, we found that female pediatricians recommended wider food variety earlier, including common food allergens, and foods considered to be triggers of symptoms in children with atopic dermatitis, such as kiwi, strawberry, and tomato [35,60,61]. On the other hand, pediatricians with more years of practice recommended fresh fruits and white rice earlier, in line with past standard recommendations [62].

Pediatricians’ decisions regarding CF are undoubtedly influenced by their own family habits [63,64], traditional approaches [65,66], and new trends, including BLW weaning [7,24,67,68]. Parenthood was correlated with earlier recommendation by the pediatricians to introduce cow’s milk, delay introduction of wheat products, and implement BLW. BLW was an early feeding method, more often promoted by pediatricians with fewer years of practice and non-parents. Although BLW has been proposed as a method for reducing food fussiness and increasing food enjoyment and satiety responsiveness [69], it should be accompanied by appropriate education of the parents, as it may increase the risk of dietary deficiencies in important nutrients, such as zinc [70], or pose a choking hazard [71].

## 5. Limitations

The current study has certain limitations. The data are based on self-reported responses, a mode of assessment that can increase response bias, which we tried to minimize by ensuring the anonymity of the respondents. The response rate was 17.4%, with only 11.65% fully completed questionnaires. This rate was similar to that in other research studies conducted by distributing the questionnaire via e-mail [27,72], but carries the risk of respondent bias. Finally, due to the nature of our study, which focused on general recommendations, specific information pertaining to individual infants and their interactions with pediatricians was not collected. Subsequent research could explore the specific guidance provided in a dyad mode, investigating the nuanced interactions between pediatricians and individual infants to enhance our understanding of tailored recommendations.

## 6. Conclusions

This study of pediatricians revealed that, in their routine practice, they appear to follow a food introduction schedule when providing guidance for CF of infants. They tend to delay the introduction of known food allergens, and to recommend a longer time period between introduction of new foods for infants at high risk of allergy. In view of the reported benefits of healthy CF for infants at both low and high risk of allergy, pediatricians should keep well informed of research outcomes and implement relevant guidelines effectively.

## Figures and Tables

**Figure 1 nutrients-16-00239-f001:**
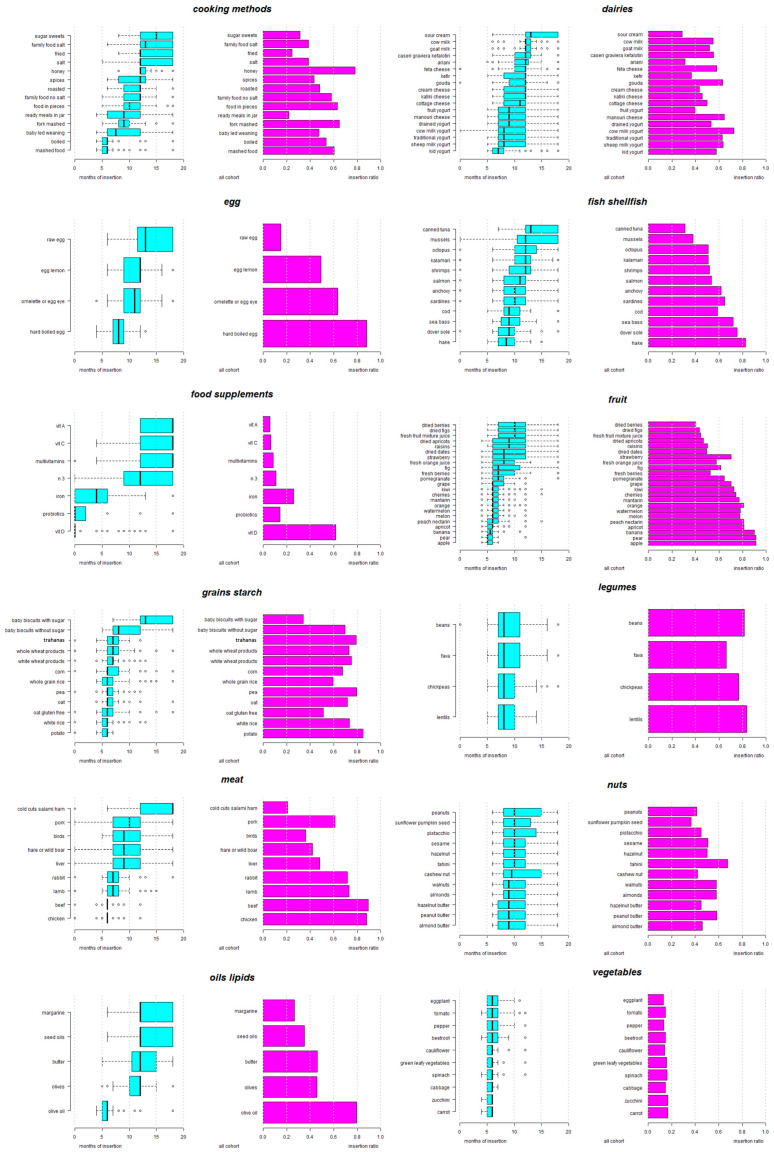
Usual guidance on complementary feeding provided by pediatricians. Months of age that the study pediatricians usually recommend for the insertion of the respective food into the infant’s diet. Insertion ratio represents the rate of respondents that selected the specific food in relation to the overall study pediatricians.

**Table 1 nutrients-16-00239-t001:** Month of age of recommendation of complementary food introduction according to the years of pediatric experience.

Food	Low Pediatric Experience (<15 Years) N = 112 Month of Age	High Pediatric Experience (≥15 Years) N = 121 Month of Age	*p*-Value
Raisins	10 (7–12)	8 (6–12)	0.048
Fresh mixed fruit juice	12 (8–12)	8 (6–12)	0.004
White rice	6 (6–7)	6 (5–6)	0.002
Rabbit	6 (6–7)	7 (6–8)	0.035
Lamb	6 (6–7)	7 (6–9)	0.007
Beef	6 (6–6)	6 (6–7)	0.024
Pork	9 (7–12)	12 (8–12)	0.036
Omelet or fried egg	10 (8–12)	12 (10–12)	0.019
Raw egg	18 (12–18)	12 (9–13)	0.009
Cow’s milk	12 (12–12)	12 (12–18)	0.038
Traditional yogurt	7 (7–10)	10 (7–12)	0.022
Sheep’s milk yogurt	7 (7–9)	9 (7–12)	0.024
Manouri cheese	8 (7–12)	10 (8–12)	0.047
Beans	8 (7–10)	9 (8–12)	0.019
Lentils	8 (7–9)	9 (7–10)	<0.001
Chickpeas	8 (7–10)	9 (7–12)	0.002
Anchovy	9 (8–12)	12 (8–12)	0.031
Calamari	12 (9–12)	13 (12–18)	<0.001
Octopus	12 (8–12)	13 (12–18)	<0.001
Shrimps	12 (8–12)	12 (12–15)	<0.001
Mussels	12 (9–13)	13 (12–18)	0.001
Omega-3	13 (12–18)	4 (0–11)	<0.001
Multivitamins	18 (18–18)	9 (4–13)	0.033
Vitamin C	18 (15–18)	6 (5–9)	0.009

Continuous variables were expressed as median (interquartile range), and comparisons were conducted utilizing the non-parametric Mann–Whitney U test. Statistical significance *p* < 0.05.

**Table 2 nutrients-16-00239-t002:** Month of age of recommendation of complementary food introduction according to the sex of the pediatrician.

Food	Male N = 56	Female N = 177	*p*-Value
Fig	6 (6–10)	8 (6–12)	0.048
Kiwi	6 (5–6)	6 (6–8)	0.003
Pomegranate	6 (5–8)	7 (6–9)	0.023
Raisins	6 (6–10)	10 (7–12)	0.006
Soft egg, omelet or fried egg	12 (10–12)	10 (9–12)	0.025
Fruit yogurt	7 (6–12)	11 (8–12)	0.042
Kids’ yogurt	7 (6–7)	7 (6–8)	0.037
Katiki cheese	10 (8–12)	12 (8–12)	0.006
Ariani	12 (6–12)	12 (12–13)	0.049
Olive oil	6 (6–6)	6 (5–6)	0.001
Baby biscuits with sugar	12 (11–14)	13 (12–18)	0.044
Boiled food	6 (6–6)	6 (5–6)	0.014
Mashed food	6 (6–8)	6 (5–6)	0.002

Continuous variables were expressed as median (interquartile range), and comparisons were conducted utilizing the non-parametric Mann–Whitney U test. Statistical significance *p* < 0.05.

**Table 3 nutrients-16-00239-t003:** Regression model of the factors that contribute to earlier recommendation by the pediatrician of introduction of specific foods in the infant’s diet.

	*p*-Value	OR	95% CI *	
**Experience (<15 years)**				
Orange	0.02	0.32	0.12	0.86
Kiwi	0.04	0.44	0.20	0.95
Strawberry	0.04	0.44	0.20	0.95
Hard-boiled egg	0.04	0.53	0.29	0.98
Wheat (gluten) products	0.007	0.35	0.17	0.75
Frumenty	0.02	0.51	0.29	0.91
Cow’s milk yogurt	0.02	0.43	0.21	0.87
Peanuts/peanut butter	0.001	0.21	0.09	0.52
Hazelnut butter	0.03	0.31	0.11	0.89
Pistachios	0.04	0.42	0.18	0.94
Almonds/almond butter	0.003	0.23	0.08	0.61
Sesame/tahini	<0.001	0.15	0.05	0.40
Beans	0.01	0.17	0.04	0.64
Seafood	<0.001	0.18	0.08	0.42
**Location (rural/semi-urban)**				
Almonds/almond butter	0.04	0.43	0.19	0.96
Walnuts	0.04	0.48	0.23	0.98
Sesame/tahini	0.04	0.46	0.22	0.97
Hazelnuts/hazelnut butter	0.02	0.33	0.12	0.85
Seafood	0.009	0.06	0.01	0.51
Beans	0.001	0.16	0.05	0.49
Lentils	0.01	0.07	0.01	0.54
Tomato	0.03	0.07	0.01	0.80
**Subspecialty (No)**				
Orange	<0.001	0.18	0.08	0.40
Kiwi	<0.001	0.29	0.15	0.56
Strawberry	<0.001	0.29	0.15	0.56
Hard-boiled egg	0.03	0.56	0.33	0.95
Soft egg/omelet/fried egg	0.004	20.49	10.32	40.66
Cream cheese	0.03	20.28	10.08	40.81
Gouda cheese	0.04	0.49	0.25	0.96
Almonds/almond butter	<0.001	0.22	0.10	0.48
Walnuts	0.01	0.40	0.20	0.81
Beans	0.002	0.23	0.09	0.58
Lentils	0.001	0.08	0.02	0.38
Seafood	0.002	0.10	0.02	0.43
**Parenthood (No)**				
Cow’s milk	0.04	0.26	0.07	0.95
Wheat (gluten) products	0.03	3.27	1.14	9.37
**Sex (Male)**				
Gouda cheese	0.02	0.24	0.06	0.83
Wheat (gluten) products	0.01	0.15	0.03	0.67
Kiwi	0.01	0.15	0.03	0.67
Strawberry	0.01	0.15	0.03	0.67

* 95% CI of the OR. OR, odds ratio; CI, confidence intervals.

**Table 4 nutrients-16-00239-t004:** Regression model of the factors that contribute to a longer time period (>4 days) between the introduction of new foods in the infant’s diet.

Time between New Foods		*p*-Value	OR	95% CI *	
Healthy infants (high)	Sex, male	0.04	1.89	1.03	3.47
Infants at risk for allergy	Subspecialty (no)	<0.001	5.91	3.05	11.43
	Location (rural)	0.002	2.90	1.47	5.70

* 95% CI of the OR. OR, odds ratio; CI, confidence intervals.

## Data Availability

Data is contained within the article (and Appendix A).

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
