# Peer review of "Complementary Feeding Practices: Recommendations of Pediatricians for Infants with and without Allergy Risk"

_nutrients, 2024, doi:10.3390/nu16020239_

Round 1

Reviewer 1 Report

Comments and Suggestions for Authors

The article provides valuable and insightful recommendations for complementary feeding practices, catering to both infants with and without allergy risk. The guidelines offered by pediatricians are practical and easy to understand, making it a helpful resource for parents navigating the complex world of introducing solid foods.

Given the dynamic nature of healthcare recommendations, it would be beneficial to encourage regular updates to the article. This ensures that the information remains current and aligns with the latest developments in pediatric nutrition.

Author Response

Thank you for your positive encouraging comments on our work. Given the new knowledge gained from studies in the area of complementary feeding, we will try to follow up and update our work.

Reviewer 2 Report

Comments and Suggestions for Authors

The paper "Complementary Feeding Practices: Recommendations of 2 Pediatricians for Infants with and Without Allergy Risk" by Vassilopoulou et al. is a survey of Greek pediatrician recommendations for babies in the presence or absence of allergy risk.

Although statistical analyses have been well conducted the opinion of this reviewer is that the experimental design lacks in terms of measuring data on babies/children. 

It would be really interesting to add allergenic measurements and related data on babies. Although I understand the rationale behind the paper analysis workflow, no scientific evidence corroborated the given pediatrician recommendations in terms of biochemical/clinical measurements i.e. IgE and other antigenic markers. 

Author Response

Thank you for your comment. Your suggestion is intriguing and could serve as the foundation for a new study examining specific guidance within a dyad, i.e., the interaction between the pediatrician and a particular infant. Our current study focused on investigating general recommendations, and unfortunately, we did not collect specific information on individual infants. We have incorporated your insightful suggestion into our study's limitations and future research.

Reviewer 3 Report

Comments and Suggestions for Authors

Dear Authors,

I thank the Editor for entrusting me to review this manuscript. I commend the authors for undertaking the work on complementary feeding practices in children with and without allergy risk. The knowledge gained from pediatricians seems to be very valuable. It is pediatricians who often encounter the problem of allergy in children, moreover, the scope of their interest is also the source of its formation.

Below are my suggestions / comments:

Despite the Author's opinion that this is a weakness of the study, in my opinion this is a good result, a 233-person sample of pediatricians included in the survey.

Table1 and 2: What do the values in parentheses represent? Why are there statistically significant differences for the same values? In the footer of the table this should be described. Why in Table 1, 15 years of pediatricians was the limit of the division of doctors?

Table 3 and 4: If these are Logistic Regression results, the interpretation of the results obtained is missing. The values of coefficients β, exp(β) or (OR) should be presented, whose values should be interpreted. It would also be better to present the 95%CI for OR than for β. In the footers of the table is "significance set at 95% level". These are the 95%CI (Confidence Intervals) da coefficients of the model.

Author Response

Despite the Author's opinion that this is a weakness of the study, in my opinion this is a good result, a 233-person sample of pediatricians included in the survey.

Thank you for your encouraging comment

Table1 and 2: What do the values in parentheses represent? Why are there statistically significant differences for the same values? In the footer of the table this should be described. Why in Table 1, 15 years of pediatricians was the limit of the division of doctors?

Reply:

We thank the Reviewer for raising this point.

In Tables 1 and 2, values are expressed as median and values in parenthesis represent the interquartile range. Comparisons between the continuous variables were conducted utilizing the non-parametric Mann-Whitney U test; although medians were similar between some variables, there were statistically significant difference due to the differences in the interquartile range. The 15 years of experience was an arbitrary cut-off we used to divide pediatricians based on their clinical experience, while similar cut-offs were used by other research studies (Choudhry NK, Fletcher RH, Soumerai SB. Systematic review: the relationship between clinical experience and quality of health care. Ann Intern Med. 2005 Feb 15;142(4):260-73. doi: 10.7326/0003-4819-142-4-200502150-00008. PMID: 15710959).   

In the revised manuscript we have included a footnote in Tables 1 and 2.

The following text has been added in the revised manuscript under the Tables 1 and 2:

‘Continuous variables were expressed as median (interquartile range), and comparisons were conducted utilizing the non-parametric Mann-Whitney U test.’

Table 3 and 4: If these are Logistic Regression results, the interpretation of the results obtained is missing. The values of coefficients β, exp(β) or (OR) should be presented, whose values should be interpreted. It would also be better to present the 95%CI for OR than for β. In the footers of the table is "significance set at 95% level". These are the 95%CI (Confidence Intervals) da coefficients of the model.

We thank the Reviewer for raising this point.

Tables 3 and 4 present the Logistic Regression of the factors that contribute to earlier recommendation by the pediatrician of introduction of specific foods in the infant’s diet (Table 3) and the factors that contribute to a longer time period (>4days) between the introduction of new foods in the infant’s diet (Table 4). Under column ‘b’ we have presented the Odds Ratios (OR) for each variable of the analysis and under ‘95% CI’ the 95% CI (Confidence Intervals) for the OR. For consistency, values of OR and 95% CI have been depicted with up to two decimals.

In the revised manuscript we have revised the Tables 3 and 4, depicting the OR and the 95% CI of the OR, including a footnote in Tables 3 and 4.

The following text has been added in the revised manuscript under the Tables 3 and 4:

‘* 95% CI of OR. OR, Odds Ratio; CI, Confidence Intervals.’

Round 2

Reviewer 2 Report

Comments and Suggestions for Authors

After the introduction of the study limitation, the paper is ready to be published.